# Low thermal conductivity of iron-silicon alloys at Earth's core conditions with implications for the geodynamo

Wen-Pin Hsieh [1,2 ✉], Alexander F. Goncharov [3,4,5 ✉], Stéphane Labrosse [6], Nicholas Holtgrewe[4,7], Sergey S. Lobanov [4,8], Irina Chuvashova[4], Frédéric Deschamps[1] & Jung-Fu Lin [9 ✉]

Earth's core is composed of iron (Fe) alloyed with light elements, e.g., silicon (Si). Its thermal conductivity critically affects Earth's thermal structure, evolution, and dynamics, as it controls the magnitude of thermal and compositional sources required to sustain a geodynamo over Earth's history. Here we directly measured thermal conductivities of solid Fe and Fe–Si alloys up to 144 GPa and 3300 K. 15 at% Si alloyed in Fe substantially reduces its conductivity by about 2 folds at 132 GPa and 3000 K. An outer core with 15 at% Si would have a conductivity of about 20 W m$^{-1}$ K$^{-1}$, lower than pure Fe at similar pressure–temperature conditions. This suggests a lower minimum heat flow, around 3 TW, across the core–mantle boundary than previously expected, and thus less thermal energy needed to operate the geodynamo. Our results provide key constraints on inner core age that could be older than two billion-years.

[1] Institute of Earth Sciences, Academia Sinica, Nankang, Taipei 11529, Taiwan. [2] Department of Geosciences, National Taiwan University, Taipei 10617, Taiwan. [3] Key Laboratory of Materials Physics, Institute of Solid State Physics CAS, Hefei 230031, China. [4] Earth and Planets Laboratory, Carnegie Institution of Washington, Washington, DC 20015, USA. [5] Institut de Physique du Globe de Paris, Université de Paris, Paris 75005, France. [6] Univ Lyon, ENSL, Univ Lyon 1, CNRS, LGL-TPE, F-69007, Lyon, France. [7] Department of Mathematics, Howard University, Washington, DC 20059, USA. [8] GFZ German Research Center for Geosciences, Telegrafenberg, 14473 Potsdam, Germany. [9] Department of Geological Sciences, Jackson School of Geosciences, University of Texas at Austin, Austin, TX 78712-0254, USA. ✉email: wphsieh@earth.sinica.edu.tw; agoncharov@carnegiescience.edu; afu@jsg.utexas.edu

Thermal conductivity in Earth's core plays a fundamental role in controlling the dynamics and evolution of this region[1]. Core convection and the resulting geodynamo are predominantly driven by thermal and compositional sources[2–4]. Energy and entropy balances of the core indicate that a convective geodynamo requires a minimum core–mantle boundary (CMB) heat flow to operate, where the minimum value increases with increasing core thermal conductivity. If the core thermal conductivity is low enough, purely thermal convection may have sustained a geodynamo for the entire Earth history. By contrast, if thermal conductivity of the core is high, the isentropic heat flux across CMB is high and compositional convection, which lowers the value of the critical CMB heat flow, is needed to sustain a geodynamo[5–9]. For the generation of most recent magnetic fields, crystallization of the inner core provides a substantial latent heat and compositional source allowing geodynamo to operate even at a high core thermal conductivity[5]. Precipitation and transport of light elements, e.g., Si, O, Mg, etc., from the outer core to the lowermost mantle have also been proposed as possible mechanisms to run a geodynamo in ancient Earth ~3.4 Gyr ago[6–9], before the inner core started to grow. Core thermal conductivity, influenced by its exact composition and temperature over its history, thus holds a key to decipher the enigmatic thermal and compositional evolutions of Earth's core, providing important insights into the origin and history of palaeomagnetic fields, available thermal vs. compositional energy sources for driving the geodynamo, and age and growth rate of the inner core[10,11].

In the past decades, geophysical and geochemical observations have revealed density deficits in Earth's inner and outer cores. Comparison between seismic models and the density of pure Fe at relevant core pressure ($P$)–temperature ($T$) conditions suggests that a certain amount of light elements alloyed with Fe is present in the core[1,12–14]. Among candidate light elements, Si is a likely candidate with ~8 and 4 wt% (≈15 and 7 at%) in the outer and inner cores, respectively, due to its geophysical and geochemical characteristics[1,12–14]. Other light elements such as O, S, C, or H could also exist in the core with Si. Moreover, at high $P$–$T$ conditions relevant to the core, Fe–Si alloy is stable in hexagonal close-packed (hcp) structure as Si is readily dissolved in Fe, and its physical properties, e.g., sound velocities and density, are able to account for the seismic data observed in the core[1,12–14]. These features motivate us to use Fe–Si alloy as a representative to investigate effects of light elements on the thermal conductivity of Fe in the core and to access the importance of various energy sources for the geodynamo.

There are two major mechanisms of heat transfer, i.e., thermal conduction and convection, in the core, where the thermal radiation mechanism does not effectively transfer heat in metallic Fe and Fe-rich alloys. Though thermal conduction of Fe alloyed with light elements at core conditions is essential to reconstruct thermal history of the core and geodymano, it has never been directly measured at relevant high $P$–$T$ conditions. Previous theoretical calculations have predicted a highly thermally conductive core with a thermal conductivity of about 80–200 W m$^{-1}$ K$^{-1}$ at the outer core and 150–300 W m$^{-1}$ K$^{-1}$ at the inner core, respectively[15–18]. These results, however, are difficult to reconcile with observations of early magnetic fields[11,19] because these high conductivity values suggest a young inner core and require either a very hot initial core[10] or alternative buoyancy sources in the form of light element extraction from the top of the core[6–8] to explain the ancient dynamo. Recent studies[20,21] on the pure Fe thermal conductivity at the outermost core conditions using two different experimental approaches show a large discrepancy: a high value of about 226 W m$^{-1}$ K$^{-1}$ was inferred from the electrical resistivity data[21], while a low value of about 33 W m$^{-1}$ K$^{-1}$ was obtained by measurements using transient heating (TH) laser

technique[20]. These results led to contradictory implications for the age and heat flow budget of the core. Prior estimates of core thermal conductivity from experiments largely focused on converting electrical resistivity of Fe and Fe alloys at high $P$–$T$ conditions into thermal conductivity via the Wiedemann–Franz (WF) law with ideal Lorenz number[21–27], while the validity of WF law at high $P$–$T$ conditions remains uncertain[15]. As a result, direct and precise thermal conductivity measurements on Fe alloyed with a major light element at relevant high $P$–$T$ conditions are critically needed to pin down core's thermal conductivity and to correctly describe the core evolution and dynamics.

In this paper, we showed that the thermal conductivity of Fe alloyed with 15 at% Si is approximately half of the pure Fe at outer core conditions. This suggests that Earth's geodynamo could be operated by pure thermal convection and that the age of inner core could be older than two billion-years.

## Results

**Thermal conductivity at high pressure and room temperature**. We combined ultrafast time-domain thermoreflectance (TDTR) with diamond anvil cell (DAC) technique to precisely measure the thermal conductivity of both single-crystal and powder samples of pure Fe and powder of Fe$_{1-x}$Si$_x$ ($x = 0.04$ and 0.15) alloys to 120 GPa at room temperature. TDTR is a well-developed ultrafast metrology that uses sub-picosecond optical pulses to pump and probe thermal transport through the sample, providing high-precision thermal conductivity measurements at pressures over 100 GPa[28,29] (Methods). The thermal conductivity of body-centered cubic (bcc) Fe (black symbols in Fig. 1) at ambient conditions is ≈76 W m$^{-1}$ K$^{-1}$. Upon compression, the thermal conductivity increases with pressure, while drastically decreases at $P ≈ 13$ GPa due to the structural transition from bcc to hcp phase, where the enhanced electron correlation reduces lifetimes of quasiparticles and thus decreases the thermal conductivity[30]. Interestingly, the pressure dependence of thermal conductivity shows a minimum around 40 GPa, which may be associated with an electronic topological transition[31], and then increases again with pressure, reaching ≈120–130 W m$^{-1}$ K$^{-1}$ near the CMB pressures.

Compared with pure Fe, the thermal conductivity of Fe$_{0.96}$Si$_{0.04}$ alloy (blue symbols in Fig. 1) at ambient conditions is significantly reduced to 16.5 W m$^{-1}$ K$^{-1}$, much lower than the previously estimated light element effects[17,20,24]. Upon compression, the thermal conductivity increases slowly until $P ≈ 40$ GPa, after which it saturates and remains at ≈40 W m$^{-1}$ K$^{-1}$ to around 110 GPa, i.e., a factor of 3 smaller than the pure hcp-Fe at similar pressures. Moreover, addition of 15 at% Si impurity further decreases the thermal conductivity of hcp-Fe$_{0.96}$Si$_{0.04}$ at high pressure (see Fe$_{0.85}$Si$_{0.15}$, red symbols in Fig. 1). At ambient conditions the thermal conductivity starts from an even lower value of 11.5 W m$^{-1}$ K$^{-1}$; similar to Fe$_{0.96}$Si$_{0.04}$, it increases slowly with pressure, while saturates to ≈19 W m$^{-1}$ K$^{-1}$ between $P ≈ 35$–120 GPa, ~6–7 fold smaller than the pure hcp-Fe. We note that alloying 4 and 15 at% (≈2 and 8 wt%, respectively) Si in Fe substantially changes the pressure dependence of thermal conductivity (i.e., the concave behavior around 40 GPa was only observed in pure hcp-Fe, not in Fe–Si alloys), suggesting that even small Si doping may stabilize the topology of the Fermi surface of hcp-Fe under compression. The substantial suppression of the thermal conductivity with the addition of 4 and 15 at% Si in Fe is presumably due to the strongly inelastic electron-impurity scattering[22,25,27].

**Thermal conductivity at high pressure–temperature conditions**. To constrain the combined effects of silicon alloying and

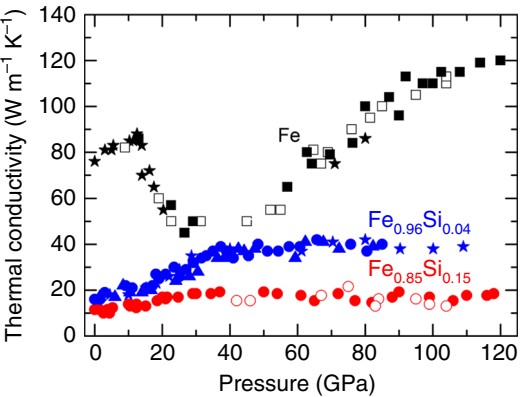

**Fig. 1 Thermal conductivity at high pressure and room temperature.** The thermal conductivity of powder Fe (black squares) is comparable with that of single-crystal Fe (black stars) and much larger than that of powder $Fe_{0.96}Si_{0.04}$ (blue symbols) and $Fe_{0.85}Si_{0.15}$ (red circles), indicating the strong alloying effect of silicon on the thermal conductivity of Fe. Each set of data includes several runs of measurement with solid symbols for compression and open symbols for decompression cycle, respectively. The measurement uncertainties are $\approx 10\%$ before 30 GPa, $\approx 20\%$ at 60 GPa, and $\approx 25\%$ at 120 GPa. The drastic decrease in the thermal conductivity of Fe around 13 GPa results from the bcc–hcp structural transition[30].

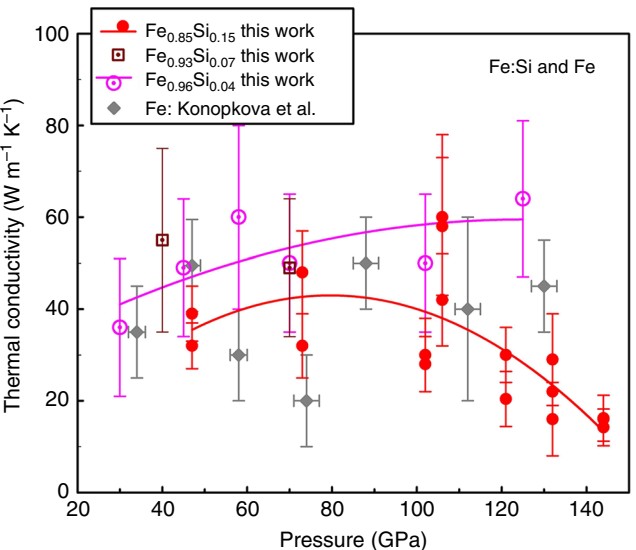

**Fig. 2 High-pressure thermal conductivity of Fe–Si alloys at 2050–3300 K.** Red and magenta curves are guides to the eye. Literature data for Fe at comparable high $P$–$T$ conditions from ref. [20] are plotted for comparison. Our results are representative of measurements with different laser powers, each corresponding to an averaging of usually 100 laser heating events using a streak camera[20,32]. The measurement uncertainties are typically $\approx 15$–$30\%$. Effects of temperature on the thermal conductivity of $Fe_{0.85}Si_{0.15}$ alloy are shown in Supplementary Fig. 4. Pressure–temperature conditions for each measurement are listed in Supplementary Tables 1–3.

high temperature, we employed the TH laser technique to measure the thermal conductivity of $Fe_{1-x}Si_x$ ($x = 0.04$, 0.07, and 0.15) at high $P$–$T$ conditions. The TH method is a well-developed pulsed laser technique to measure thermal conductivity at simultaneously high $P$–$T$ conditions[20,32], where the heat pulses across the sample are probed temporally and spatially using in situ time-domain thermoradiometry, and the thermal conductivity is deduced from the results of the model finite-element (FE) calculations (Methods). Figure 2 shows the thermal conductivity of polycrystalline Fe–Si alloys to 144 GPa at 2050–3300 K. Considering the measurement uncertainty, the thermal conductivities of hcp-$Fe_{0.96}Si_{0.04}$ and hcp-$Fe_{0.93}Si_{0.07}$ (magenta dotted circles and brown dotted squares, respectively) below ~110 GPa are comparable with the pure hcp-Fe[20]. We should note that the large scatter in the literature for pure Fe data[20] around 40–90 GPa was assigned to be partially associated with the presence of γ phase which could affect some of its results at high $P$–$T$ conditions; however, the γ-Fe disappears above 100 GPa, so the data scatter less in this regime. The thermal conductivity of $Fe_{0.85}Si_{0.15}$ (red circles), on the other hand, is slightly smaller than the pure hcp-Fe below 100 GPa, though their differences are within uncertainties. Importantly, the thermal conductivity of $Fe_{0.85}Si_{0.15}$ decreases significantly with increasing pressure from ~120 to 144 GPa. Furthermore, unlike pure hcp-Fe whose thermal conductivity decreases with increasing temperature (Supplementary Fig. 1), the thermal conductivity of hcp-$Fe_{0.96}Si_{0.04}$ at high temperatures is comparable or slightly larger than that at 300 K (Supplementary Fig. 2). Moreover, the thermal conductivity of $Fe_{0.85}Si_{0.15}$ at high temperatures is generally larger than at 300 K, except at the highest pressures where they become very close to each other (Supplementary Figs. 3 and 4). It is worth noting that at the high $P$–$T$ conditions of our TH experiments, $Fe_{0.96}Si_{0.04}$ stabilizes in the hcp phase[33] (Supplementary Fig. 5), whereas $Fe_{0.85}Si_{0.15}$ almost exclusively falls into the mixed hcp–bcc phase region[12] (Supplementary Fig. 6) which might result in an increase of the thermal conductivity. However, based on the 300 K data, there is no abrupt sizable change in the thermal conductivity of $Fe_{0.85}Si_{0.15}$ near 40 GPa (where this transition is expected to occur at 300 K[12]); only a change in the

pressure slope is observed so the thermal conductivity remains approximately constant (Fig. 1). We thus conclude that the observed conductivity behavior for $Fe_{0.85}Si_{0.15}$ is mainly due to Si alloying effect in the hcp phase, instead of the hcp–bcc mixture in the sample.

## Discussion

Extrapolation of our room-temperature pure hcp-Fe thermal conductivity data to relevant high-temperature conditions confirms the consistency of the TDTR results with the TH results. The TDTR data for pure hcp-Fe at the outermost core pressures and 300 K is about 120 W m$^{-1}$ K$^{-1}$ (Fig. 1). If we assume the temperature dependence of the pure hcp-Fe follows a $T^{-1/2}$ dependence as estimated by Konôpková et al.[20], the thermal conductivity of hcp-Fe at the outermost core conditions ($\sim P = 136$ GPa, $T = 3800$–4800 K) is estimated to be about 30–33.7 W m$^{-1}$ K$^{-1}$, nearly the same as that (33 ± 7 W m$^{-1}$ K$^{-1}$) obtained by TH measurements (ref. [20] and Supplementary Fig. 1). As for the Fe–Si alloys, however, the exact temperature dependence of thermal conductivity at high pressures likely varies with Si content and applied pressure (Fig. 2 and Supplementary Fig. 4), which remains relatively uncertain. Thus, it would be difficult to make unambiguous extrapolation of room-temperature TDTR data to high-temperature conditions and compare them with the high-temperature TH data. Nevertheless, we note that qualitatively these two sets of data correspond reasonably well as both sets of data demonstrate pressure dependencies with a broad maximum for Fe–Si alloys (after about 40 GPa at 300 K and around 80–100 GPa at 2050–3300 K). Moreover, given the Si alloying effect, it is expected that the Fe–Si alloys would have weaker temperature dependences than the pure hcp-Fe, since the presence of impurities will enhance the scattering of carries (phonons and electrons) during the transport of energy.

This qualitative behavior is clearly indicated in Supplementary Figs. 1–3.

We further compare our results with previous studies to disentangle the Si light element effect from the temperature-dependent thermal conductivity of Fe and Fe–Si alloys at Earth's core pressures. The thermal conductivity of metals is mainly determined by the electronic contribution, which is the case for pure Fe, where the lattice thermal conductivity is negligible[24]. However, at high pressure and room temperature, the electrical conductivities of hcp-$Fe_{0.96}Si_{0.04}$ and $Fe_{0.84}Si_{0.16}$ ($\approx$2 and 9 wt% Si, respectively) alloys, similar in composition to our samples, were found to be smaller than that of pure hcp-Fe by a factor of about 4 and 10 [22,27], respectively, due to the impurity effect. Compared with the intrinsic electron–phonon scattering, the impurity scattering effect plays a predominant role in influencing the thermal energy transport in Fe–Si alloys at high $P$–$T$ conditions. Thus, the different temperature dependence of thermal conductivity among pure hcp-Fe, $Fe_{0.96}Si_{0.04}$, and $Fe_{0.85}Si_{0.15}$ could be explained by the $T$ dependence of the impurity scattering, as doping of silicon impurity likely flattens the temperature dependence of thermal conductivity. On the other hand, the variation in high $P$–$T$ thermal conductivity of Fe–Si alloys is likely due to the $P$–$T$ effects on electron-impurity scattering contribution to the conductivity (see Supplementary Fig. 4 and Supplementary Note 1). We also note that because of a decrease in the electronic thermal conductivity contribution in Fe–Si alloys, the phonon contribution (via, e.g., the electron–phonon and phonon-impurity scatterings) to their total thermal conductivity is expected to play a non-negligible role for thermal transport[34]. The aforementioned dissimilarity in the $P$–$T$-dependent thermal conductivity makes the conductivity values of hcp-Fe and hcp-$Fe_{0.96}Si_{0.04}$ comparable with each other and about twice larger than hcp-$Fe_{0.85}Si_{0.15}$ at $P$–$T$ conditions relevant to Earth's outer core.

Prior studies reported that the electrical resistivity of solid $Fe_{0.84}Si_{0.16}$ at ~136 GPa and 3750 K, i.e., outermost core conditions, is on the order of ~$1 \times 10^{-6}$ $\Omega$ m[22,27]. Using the WF law with the ideal Lorenz number, the corresponding thermal conductivity was estimated to be about 40–60 and 90 W m$^{-1}$ K$^{-1}$ [27], respectively. If we assume Si is the major light element with $\approx$15 at% ($\approx$8 wt%) in the outer core, these high literature values of inferred thermal conductivity of solid $Fe_{0.84}Si_{0.16}$ at outermost core conditions are much larger than the $\approx$20 W m$^{-1}$ K$^{-1}$ value for solid $Fe_{0.85}Si_{0.15}$ obtained by our direct measurements (See Table 1 for a summary of recent results on the electrical resistivity and thermal conductivity of Fe and Fe–Si alloys at outer core conditions.). The large discrepancy may arise from the previously modeled temperature dependence of electrical resistivity at high pressure, or from using an assumed ideal Lorenz number at high $P$–$T$ conditions. We note that our direct thermal conductivity measurements do not involve these assumptions, yielding the robust conclusions concerning thermal evolution scenarios of the core (see geodynamic modeling below).

Our results on the thermal conductivity of solid $Fe_{0.85}Si_{0.15}$ at outer core $P$–$T$ conditions is expected to set an upper bound for that of the liquid outer core, as the extrapolation of our results (Supplementary Fig. 4) to the core temperatures (>4000 K) would not change it much, while the thermal conductivity of a material in molten phase that lacks crystallinity for heat conduction is typically smaller than in solid phase. For Fe and Fe-light element alloys, the effect of melting is expected to reduce the thermal conductivity of the solid phase by $\approx$20% or less[15,18,20–22,34–37]. For instance, Silber et al.[36] recently reported that at pressures from 3 to 9 GPa the electrical resistivity (inversely proportional to the electronic thermal conductivity using WF law) of Fe alloyed with 4.5 wt% Si abruptly increases by ~$10^{-7}$ $\Omega$ m (~10%) or less

**Table 1 Recent experimental and computational results of electrical resistivity ρ and thermal conductivity Λ of Fe and Fe–Si alloys at outer core conditions.**

| Composition | ρ (μ Ω cm) | Λ (W m$^{-1}$ K$^{-1}$) | Method | Reference |
|---|---|---|---|---|
| hcp Fe | ~90 | ~100 | C | 15 |
| hcp Fe | NA | ~33 | DTCM | 20 |
| hcp Fe | ~40 | ~226[a] | ERM | 21 |
| hcp Fe | ~60–130 | ~67–145[a] | ERM | 22 |
| Liquid Fe | ~70 | ~140 | C | 16 |
| Liquid Fe | ~70 | ~130 | C | 17 |
| hcp $Fe_{0.85}Si_{0.15}$ | NA | ~20 | DTCM | This study |
| hcp $Fe_{0.84}Si_{0.16}$ | ~150–215 | ~41–60[a] | ERM | 22 |
| hcp $Fe_{0.78}Si_{0.22}$ | ~100 | ~90[a] | ERM | 25 |
| Liquid $Fe_{0.875}Si_{0.125}$ | ~90 | ~110 | C | 17 |
| hcp $Fe_{0.65}Ni_{0.1}Si_{0.25}$ | ~112 | ~87[a] | ERM + C | 27 |

Method: C calculation, DTCM direct thermal conductivity measurement, ERM electrical resistivity measurement.
NA not applicable.
[a]Thermal conductivity was inferred from electrical resistivity via WF law.

**Table 2 Recent results on the change in electrical resistivity Δρ of Fe and Fe–Si alloys upon melting.**

| Composition | P (GPa) | Δρ (μ Ω cm) | Method | Reference |
|---|---|---|---|---|
| Fe | 329 | ~+10 | C | 18 |
| Fe | 51 | ~+20–30 | ERM | 21 |
| Fe | 12 | ~+10 | ERM | 35 |
| Fe | 5 | ~+20 | ERM | 37 |
| $Fe_{0.91}Si_{0.09}$ | 9 | ~+5 | ERM | 36 |
| $Fe_{0.82}Si_{0.18}$ | 10 | ~−50 | ERM | 38 |
| $Fe_{0.82}Si_{0.1}O_{0.08}$ | 329 | ~+15 | C | 18 |

Method: C calculation, ERM electrical resistivity measurement.

as it undergoes a solid-to-liquid transition, and such increase in resistivity is expected to be also present at higher pressures[18]. We note, however, that recent electrical resistivity data for Fe–Si alloys by Pommier et al.[38] show an opposite trend, at odds with most literature results where the electrical resistivity of metals and their alloys typically increases with temperature and upon melting[15,18,21,25,34–37] (see Table 2 for a summary of recent results on the change in electrical resistivity of Fe and Fe–Si alloys upon melting).

Assuming an outer core with 15 at% Si being the major light element, the significant reduction of Fe thermal conductivity by about twofolds caused by alloying 15 at% Si at outer core conditions, as indicated by our data, provides crucial constraints on the thermal state and geodynamo of the outer core as well as the age of the inner core. Our result for the low thermal conductivity of $Fe_{0.85}Si_{0.15}$ alloy at outer core conditions, $\approx$20 W m$^{-1}$ K$^{-1}$, represents the first direct measurement that pins down the outer core thermal conductivity to a low-end value estimated in ref. [20]. It further considerably lowers the power requirements of a thermally driven geodynamo compared with recently proposed scenarios, which in turn requires lower initial core temperatures and consequently a potentially older inner core[2,10]. More specifically, the core's thermal conductivity provides a lower bound on the power that needs to be extracted from the core at the CMB to drive a thermal geodynamo. The thermal geodynamo can obviously operate with higher power, and the real value of this power is imposed by the CMB heat flow, which is itself controlled by mantle convection and is estimated to be in the range of 5–17 TW[39]. The

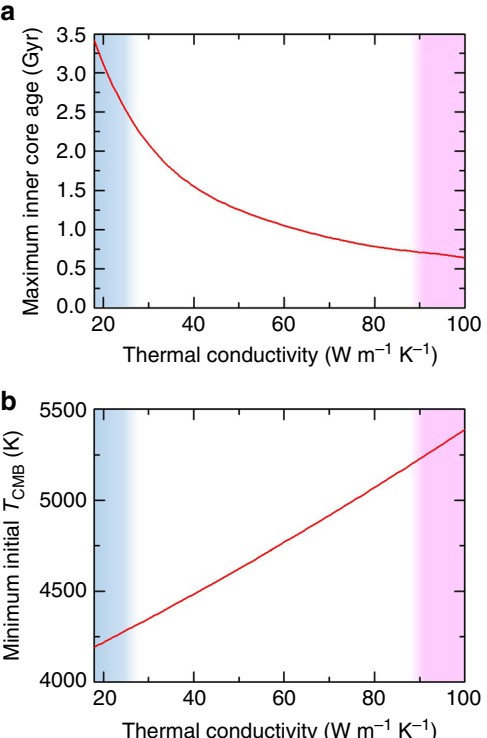

**Fig. 3 Effects of core thermal conductivity on its thermal evolution.**
**a** maximum inner core age and **b** minimum initial core–mantle boundary
(CMB) temperature as a function of thermal conductivity of the core.
Results are obtained with a thermal evolution model assuming an isentropic
CMB heat flow at each time, which is the minimum to maintain a magnetic
field by thermal convection alone. Blue shaded areas represent the range of
$Fe_{0.85}Si_{0.15}$ thermal conductivity at outer core P–T conditions indicated by
our study, while red shaded areas represent the high thermal conductivity
values of Fe–Si alloys inferred from literature results, see e.g., refs. [25,27].

thermal convection lower bound is defined by the critical CMB
heat flow at which convection turns on, i.e., the isentropic heat
flow, given by the product of thermal conductivity and isentropic
temperature gradient. The actual heat flow can be lower thanc the
isentropic value if the compositional convection occurs, owing to
the inner core growth[5] or light element extraction[6–9], which is a
tenet of buoyancy-driven dynamos[7,40]. For a thermal conductivity
of ≈20 W m$^{-1}$ K$^{-1}$, the minimum heat flow is ≈3 TW[41], i.e.,
smaller than the lower bound of an estimated modern CMB heat
flow[39], while for conductivities larger than ≈115 W m$^{-1}$ K$^{-1}$, the
minimum heat flow is larger than the upper estimate of a modern
CMB heat flow. To illustrate the key role played by the thermal
conductivity on the core evolution, we further calculated the
maximum inner core age and minimum initial CMB temperature
for a wide range of core thermal conductivity (Fig. 3). For sim-
plicity, we consider that the dynamo before the inner core
nucleation runs on heat alone, i.e., no other source than thermal is
available at that time. We computed the thermal evolution of the
core for a CMB heat flow always equal to the isentropic value, i.e.,
the minimum requirement to initiate thermal convection and run
a dynamo. The high-end values (>90 W m$^{-1}$ K$^{-1}$, red shaded areas
in Fig. 3) obtained by previous theoretical predictions[16,17] and
electrical resistivity measurements[25,27] combined with calculations
using the WF law with ideal Lorenz number result in unrealistically
high CMB temperature in the early Earth[10]. By contrast, the value
obtained by our direct thermal conductivity measurements
(≈20 W m$^{-1}$ K$^{-1}$, blue shaded areas in Fig. 3) leads to reasonable
bounds on thermal evolution scenarios.

Our results further indicate that Earth's dynamo could have
been running on the thermal energy alone throughout its history
with the additional help from compositional buoyancy when the
inner core started to crystallize. The low thermal conductivity
(≈20 W m$^{-1}$ K$^{-1}$) of the outer core enables a purely thermally
driven dynamo with an initial CMB temperature on the order
of 4500 K, which is a geochemically acceptable value[42] from
the standpoint of core formation. The significant reduction of
Fe thermal conductivity due to the Si impurity effect could be
general for other candidate light elements in the core. Addi-
tional direct high P–T thermal conductivity measurements on
O–, S–, and C–bearing binaries and more realistic ternary Fe-
light element systems are required to precisely quantify the
role played by these elements. These future studies could
strengthen the conclusion that, due to the low core thermal
conductivity, the geodynamo can run on heat alone for the
entire age of the Earth without the help of compositional
convection.

## Methods

**Starting materials and sample preparation.** Single crystals of pure Fe for TDTR
experiments at 300 K were synthesized by Princeton Scientific Corporation, Prin-
ceton, NJ. At ambient conditions, the pure Fe is in bcc phase with (100) orienta-
tion. Powder samples of pure Fe and chemically homogeneous $Fe_{1-x}Si_x$ ($x = 0.04$
and 0.15) alloys were from Goodfellow Corporation, and their crystal structures
were also in bcc phase characterized by X-ray diffraction. The chemical compo-
sition of each alloy was confirmed to be $Fe_{0.96}Si_{0.04}$ and $Fe_{0.85}Si_{0.15}$ by an electron
microprobe[12]. Before being loaded into the high-pressure DAC, the single-crystal
samples were cut to ≈50 × 50 μm$^2$ and a thickness of ≈30 μm using focused ion
beam (FIB) in Center for High Pressure Science and Technology Advanced
Research (HPSTAR), Shanghai, and powder samples were pressed to a disk shape
with a diameter of ≈50 μm and a thickness of ≈10 μm.

The $Fe_{0.85}Si_{0.15}$ alloy was synthesized in an end-loaded 150-ton piston-cylinder
press at Institut de physique du globe de Paris, by equilibrating molten metal with
molten silicate at fixed temperature and oxygen fugacity. For this, natural fresh
MORB from the mid-Atlantic ridge (GRA-N16-6) was ground and mixed with Fe
and FeSi, then fully melted and equilibrated at 2 GPa and 1800 °C for 120 s, using
an MgO capsule, a graphite furnace and $BaCO_3$ cell. After quench, the metal had
fully coalesced to a spherical ball surrounded by silicate glass; its homogeneity and
composition were analyzed by a scanning electron microscope, and it was then
crushed for loading in DAC experiments. All the samples for high-temperature TH
measurements are polycrystalline.

To measure the thermal conductivity at high pressures and 300 K, the samples
were then coated with ≈80-nm thick Al film and loaded, together with a ruby ball,
into a symmetric DAC with a culet size of 200 or 300 μm and a Re gasket. Silicone
oil (CAS No. 63148-62-9 from ACROS ORGANICS) was used as the pressure
medium. The pressure was determined by fluorescence spectrum of the ruby[43] with
a typical uncertainty of less than 5%.

In TH experiments to high temperature, the samples thinned down (by
squeezing between two diamonds) to ~4 μm were loaded in a symmetric DAC
using KCl as a pressure medium and thermal insulator. The sample position and
thickness and the distances between the sample surface and diamond tips were
measured at high pressure using optical spectroscopy of the interference fringes
recorded in the reflectivity spectra from the cavity without the sample and from the
sample surfaces from both sides[20,32]. The refractive index of KCl was determined
by extrapolating linearly the results as a function of density in ref. [44] to higher
pressures. Pressure was determined from the position of the Raman edge of the
stressed diamond anvil tip[45].

**Thermal conductivity at high pressure and room temperature.** Thermal con-
ductivities of Fe and Fe–Si alloys at high pressure and room temperature were
measured using an ultrafast optical pump-probe method, TDTR. In our TDTR
measurements, the output of a Ti:sapphire oscillator laser was split into pump and
probe beams. The pump beam heated up the Al film coated on the sample and
created temperature variations. The resulting optical reflectivity change of the Al
film as a function of time was measured by the probe beam that was delayed by
passing through a mechanical stage. The in-phase $V_{in}$ (real part) and out-of-phase
$V_{out}$ (imaginary part) components of the variations of the reflected probe beam
intensity, synchronous with the 8.7 MHz modulation frequency of the pump beam,
were detected by a Si fast photodiode and an RF lock-in amplifier. Detailed
descriptions of the TDTR method are discussed elsewhere, see, for example,
refs. [46,47].

To determine the thermal conductivity of the sample, we compared the ratio
$-V_{in}/V_{out}$ as a function of delayed time between pump and probe beams to thermal
model calculations that take into account heat flow into the sample and into the
pressure medium silicone oil[48,49]. Example data for hcp Fe at high pressures along

with calculations by the thermal model are shown in Supplementary Fig. 7. There are several parameters in our thermal model, including laser spot size, thickness of Al film, thermal conductivity, and heat capacity of each layer, but the thermal conductivity of the sample is the only significant unknown and free parameter to be determined. Under our experimental geometry and conditions, the ratio $-V_{in}/V_{out}$ during the delay time of few hundred picoseconds is most sensitive to and scales with sum of the thermal effusivity of the sample and silicone oil divided by the heat capacity per unit area of the Al film, see ref. [50] for details. The Al thickness at ambient pressure was measured in situ by picosecond acoustics; we estimated the changes in Al thickness at high pressures following a method developed in ref. [51]: Al thickness decreases by 7.8% at 25 GPa, by 10.3% at 40 GPa, by 13.1% at 70 GPa, and 15.4% at 120 GPa. In addition, at the modulation frequency of the pump beam (8.7 MHz), the thermal penetration depths in Fe, Fe–Si alloy, and silicone oil are of the order of hundreds of nanometers[52], and therefore our thermal model calculations are insensitive to their thicknesses (~10 μm), see Supplementary Fig. 8a, b. Since the Al thermal conductivity at ambient pressure is large (≈200 W m$^{-1}$ K$^{-1}$)[50] and has minimal effects on the thermal model calculations (Supplementary Fig. 8c), we fixed this value at high pressures. We estimated the Al heat capacity at high pressures from literature data for the atomic density and elastic constants at high pressures along with calculations of Debye temperature, see ref. [52] for details. The pressure dependent thermal effusivity, square root of the product of thermal conductivity and volumetric heat capacity, of silicone oil to 24 GPa was taken from ref. [53]; the thermal effusivity from 24 to 120 GPa was estimated by extrapolation of the lower pressure data that were fitted into a polynomial, assuming the silicone oil remains in an amorphous phase at these pressures. Note that the thermal effusivity of silicone oil at high pressures is much smaller than that of the Fe and Fe–Si alloys, which significantly reduces the uncertainty of the measured thermal conductivity of the sample; the exceptionally low thermal effusivity of silicone oil has minor influences on the thermal model calculations, typically less than 5% uncertainty, see Supplementary Fig. 8d.

The volumetric heat capacity of the bcc Fe at ambient pressure and room temperature is 3.54 J cm$^{-3}$ K$^{-1}$, and its pressure dependence is taken from the results of ref. [54] along with the equation of state (EOS) from ref. [55], where the relatively small electronic contribution to the heat capacity is further reduced at high pressures. For the hcp Fe, the lattice contribution to the heat capacity was taken from the results by Murphy et al.[56]. Though its electronic contribution is not well known, Wasserman et al.[57] showed that, for fcc Fe, the electronic contribution to the heat capacity is much smaller than the lattice contribution, in particular at room temperature and higher pressures. We assumed the hcp Fe has similar property as suggested by ref. [58] and thus its lattice heat capacity is predominantly and reasonably represents the total heat capacity of hcp Fe at room temperature and high pressures.

On the other hand, the volumetric heat capacities of the Fe$_{0.96}$Si$_{0.04}$ and Fe$_{0.85}$Si$_{0.15}$ alloys at room temperature and high pressures are not known. We first estimated their heat capacities at ambient conditions to be 3.72 and 4.22 J cm$^{-3}$ K$^{-1}$, respectively, by interpolating the ambient heat capacities between pure bcc Fe and FeSi[59] for 4 and 15 at% of Si. We then assumed both the Fe$_{0.96}$Si$_{0.04}$ and Fe$_{0.85}$Si$_{0.15}$ have a similar pressure dependence to that of the FeSi as calculated in ref. [59]. Since the electrical resistivities of Fe$_{0.96}$Si$_{0.04}$ and Fe$_{0.85}$Si$_{0.15}$ are larger than Fe, their total heat capacity is predominantly determined by the lattice contribution. Finally, by evaluating the sensitivity of the thermal model to input parameters, we calculated the uncertainty in the thermal conductivity of Fe and Fe–Si alloys resulting from the uncertainty in each of the parameters used in our thermal model (see, for example, refs. [50,60] for details of the uncertainty evaluation, and example tests in Supplementary Fig. 8). Importantly, precise determination of the Fe and Fe–Si alloys thermal conductivity requires the model to have higher sensitivity to their thermal conductivity but lower sensitivity to other input parameters. We found that uncertainties in all the parameters propagate to ≈10% error in the measured thermal conductivity before 30 GPa, ≈20% error at 60 GPa, and ≈25% error at 120 GPa.

**Thermal conductivity at high pressure and temperature**. Thermal conductivity at high pressure and high temperature was measured by the TH technique similar to those reported in refs. [20,32]. In our experiments, the bulk of a several μm thick sample preheated by continuous-wave lasers from both sides is probed by launching a thermal wave created by sending a microsecond (μs) long pulse from one sample side and recording its temperature history via a time resolved spectroradiometry from both samples sides (Supplementary Fig. 9). These temperature evolutions were approximated by two-dimensional (axially symmetric) FE model calculations using the experimentally determined sample geometry[20,32,61]. The EOS of Fe$_{0.85}$Si$_{0.15}$ is from ref. [12], and the EOS of KCl is from ref. [62]. The two major parameters to fit the data are the thermal conductivity of the sample and the medium (KCl). The error bars are estimated as combined uncertainties of fitting, input material and geometrical parameters, and other assumptions (e.g., neglecting thermal expansion).

## Data availability
All data supporting the findings of this study are available within the paper or available from the corresponding authors upon request.

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

## Acknowledgements
The work by W.P.H. was supported by the Academia Sinica and the Ministry of Science and Technology of Taiwan, Republic of China, under Contract AS-CDA-106-M02, 106-2116-M-001-022, and 107-2628-M-001-004-MY3. W.P.H. acknowledges the fellowship from the Foundation for the Advancement of Outstanding Scholarship, Taiwan. The work at Carnegie was supported by the NSF (grant numbers DMR-1039807, EAR-1520648, EAR/IF-1128867, EAR-1763287, EAR-1901813), the Army Research Office (grant number W911NF-19-2-0172), the Deep Carbon Observatory, the National Natural Science Foundation of China (grant number 21473211, 51672279, 11874361, and 11774354), the Chinese Academy of Science (grant number YZJJ2020QN22 and YZ201524) and the CASHIPS Director's Fund (Grant No. YZJJ201705). S.S.L. was partially supported through the Helmholtz Young Investigators Group CLEAR (VH-NG-1325). J.F.L. acknowledges support from the National Science Foundation (EAR-1901801) and the Deep Carbon Observatory. The authors acknowledge Dr. James Badro for his help with the sample preparation, and Dr. Youjun Zhang for cutting single-crystal samples using a focused ion beam system in the Center for High Pressure Science and Technology Advanced Research.

## Author contributions
W.P.H., A.F.G. and J.F.L. conceived and designed the project. W.P.H., A.F.G., N.H., S.S.L. and I.C. conducted experiments and analyzed data. S.L. performed core evolution models. W.P.H., A.F.G., S.L., F. D. and J.F.L. wrote the manuscript. All authors reviewed and commented on the manuscript.

## Competing interests
The authors declare no competing interests.
