## [Peer Review File · Nature Communications]

Reviewers' comments:

Reviewer #1 (Remarks to the Author):

The authors present the results of experiments to measure the thermal conductivity of solid iron alloys at high pressure and temperature. Two techniques are employed. The first technique is used to measure thermal conductivity at high pressure and room temperature. The second technique explores the simultaneous affects of high pressure and temperature.

The results are both important and provocative. Recent theoretical calculations have suggested that the thermal conductivity of iron alloys is quite high. There is some experimental support for these values, but the measurements are not direct estimates of thermal conductivity. Instead, measurements of electrical conductivity are related to thermal conductivity using the Wiedemann-Franz Law. A relationship between thermal and electrical conductivity is quite reasonable, although there has been some debate about the numerical constants used to relate these transport quantities.

The results are provocative because the author advocate for a much lower thermal conductivity than that obtained from theory. This reduction in conductivity has enormous consequences for the thermal evolution of the core and the origin of the Earth's magnetic field. The authors have done a good job conveying the significance of their experiments and providing insightful and quantitative estimates for the implications.

The most important question is whether the experimental measurements and analysis are sufficient to reliably support the authors' conclusions. The authors describe their efforts to propagate uncertainty through the analysis of their measurements. This is important because the thermal conductivity must be extracted from a modelling procedure. Uncertainties in the model parameters will affect the outcome. The authors suggest that the uncertainty is about 10-20%. I wonder if these error estimates are too low. Figure S4 shows the temperature dependence at various pressures. I would be surprised is the erratic pressure dependence is real, which suggests that these variations are reflection of experimental uncertainty. Still, the results are sufficient to argue for lower thermal conductivity, even with larger experimental uncertainties.

There is little doubt that this study will motivate future work, either to reproduce these results or to extend the experiments to other light elements. It seems likely that silicon will be present at some level in the core, so the results should be relevant moving forward. I would encourage the authors to make sure that their error estimates will stand the test of time.

The present study could be published in its current form, although I would encourage the authors to be clear early in the paper that the experiments are carried out on solid samples. This detail emerges once the experiments are described, but it is sufficiently important to mention in the abstract. Allowing for the likely changes due to melting only reinforces the authors' conclusions.

Reviewer #2 (Remarks to the Author):

This paper report the results of the direct determination of thermal conductivity of FeSi alloy at high pressure, and revealed that it is incorporation of Si in hcp-Fe significantly reduces the thermal conductivity of 20 Wm⁻¹K⁻¹. Which implies the heat flux at the CMB reduces to around 3 TW.

There is a discrepancy of a high thermal conductivity inferred from the electrical conductivity, whereas a low thermal conductivity from the transient heating laser method. These discrepancies are very

significant and provide the different views for the formation of the inner core and the early magnetic field.

The senior author is an expert of the thermal conductivity measurements at room temperature by using TDTR technique and the results are considered to be reliable. The thermal conductivity at high pressure and high temperature was made by the TH method, which is also considered to be reliable since the co-authors have significant experience of the measurements.

This is a clear and concise paper showing lower thermal conductivity of the Fe-Si alloys. Discussion is clear and arguments and conclusion is reasonable. Implications for the formation of the inner core and the core dynamo is very important. Therefore this paper can be suitable for this journal. However, there are so many incomplete figures and captions. Therefore, the authors should rewrite the figures and the captions completely. We cannot accept it in the present version. Major revision is needed for publication.

Specific comments especially on the figures.

Page 2, First paragraph and Page 6 the first paragraph of discussion: I would like to make some discussions on the mechanism of heat transfer at very high pressure and temperature. Generally heat is transferred by three mechanisms, such as convection heat transfer, thermal conduction which is measured here experimentally, and radiation. I am wondering that the radiation heat transfer may not be negligible and this mechanism might be important and can increase the thermal conductivity significantly in spite of the low thermal conductivity determined here. We need some discussions on the heat transfer mechanism in the core.

Page 5, 2nd paragraph: The authors used two techniques, TDTR at 300 K and TH at high temperature. However, there is no discussion on the consistency of these technique. I would like to see some discussions on the consistency of the two data-sets. The data on TDTR at room temperature might extrapolate theoretically, and could compare the data by TH. The Si content dependency for TDTR and TH are consistent?

Figure S3: There is no explanation of the blue curve. No explanation for FE (finite-elements) calculation. FE calculations should be FE calculation. Please fix this figure.

Figure S4: The change of the FeSi thermal conductivity shows strange pressure dependency at high temperature, i.e. there is a maximum at 106 GPa. Provide a reasonable explanation of the pressure dependency at high temperature of 2500-3000 K. Is it due to a two phase mixture of BCC and HCP? If the anomalous change is due to uncertainty of the TH method, please provide uncertainty for the plot for the measurements.

Figure S5: Phase diagram shows Fe 2-3.5wt% Si. In the caption, alloy compositions are given as atomic in composition. Fe 2-3.5wt%Si should change to atomic ration as shown in the caption. It is confusing.

The symbols of this figure are not clear. There are blue diamonds and upwards triangle, blue open circles and downwards triangles, and orange open stars. There are no detailed explanations of the symbols. What does the meaning of the top side of the sample and the bottom side of the sample? The top side and bottom side of the sample is related to the pulsed side and opposite side given in Figure S3?

Figure S6: Provide a reference for the phase diagram of FeSi alloy. It may be ref. 12, but it is confusing that black triangles and open circles are the literature data for phase identification by ref. 12,

but NOT the data points for previous thermal or electrical conductivity measurements. The measurement made with TH experiments was for the mixture of hcp and bcc, and not a single phase. Please explain the differences of your data points shown in yellow stars and brown stars. Do you show the two series of measurements?

What is "Bottom temperature" in this figure?

The solid lines for the boundaries of bcc, hcp, fcc may be those of pure Fe instead of Fe_{0.85}Si_{0.15}. Therefore it should be deleted.

Reviewer #3 (Remarks to the Author):

Review on "Low thermal conductivity of iron-silicon alloys at Earth's core conditions: implications for the geodynamo" by Hsieh et al.

In this study, the authors measured thermal conductivity of Fe and Fe-Si alloys at high pressures and room temperature and at high pressures and high temperatures by means of the transient heating method in DAC. They report that the thermal conductivity of Fe-Si alloy is lower than pure Fe at high temperatures and ambient temperature. They also report that thermal conductivity of 20-60 W/mK at high pressures and temperatures, which is much lower than previously estimated using electrical conductivity based on the Wiedemann-Franz's law. According to these results, they concluded that the inner core is older and that the initial core-mantle-boundary is colder than previously estimated.

I have several scientific comments on this manuscript.

1. I think that their main conclusion that the Fe-Si alloys have low thermal conductivity than pure Fe is invalid at high pressures and temperatures. Figure 2 demonstrates that the thermal conductivities of pure Fe and Fe-Si alloys are simply indistinguishable at high temperatures, although it may be the case at ambient temperature as shown in Fig. 1.
2. As shown in Table S1-S3, the measurements for Fe_{0.96}Si_{0.04} and Fe_{0.93}Si_{0.07} were conducted at a temperature of 2050 K, whereas those for Fe_{0.85}Si_{0.15} were at temperatures of 2300 to 3100 K. Therefore, this study does not present consistent experimental results to show the compositional dependence of thermal conductivity of Fe-Si alloys.
3. The thermal conductivity of Fe and Fe-Si alloys increases by adding percent orders of Si to Fe, whereas it decreases by adding 15at% of Si. This would indicate that the effect of Si addition is monotonic, and therefore, they cannot argue core thermal conductivity simply using that of Fe_{0.85}Si_{0.15} alloy.
4. Figure S4 shows that the thermal conductivity of Fe_{0.85}Si_{0.15} increases with increasing temperature at pressures of 47~106 GPa, whereas it is independent from temperature at higher pressures. The author need to argue the mechanism to cause these phenomena.
5. The outer core temperatures are more than 4000 K. The present temperature range is insufficient for arguing the core conductivity, because the temperature effect on thermal conductivity does not seem simple as argued above.
6. I doubt that argument of the core thermal history based on thermal conductivity of core materials is essential. The core has cooled due to heat release across the CMB. If there were no heat production in the mantle, the released heat should be equal to the total surface heat flow. Thus, the core cooling-rate should be controlled by the surface heat flow of the Earth not by the heat transfer in the core. This argument is strengthened by the fact that the thermal conductivity of rocks is much lower than that of Fe alloys. Therefore, it is the thermal-boundary-layer structure of the bottom of the Earth that controls the amounts of heat released from the core to the mantle. The amounts of heat transfer

within the core should be adjusted by verbosity of thermal convection in the outer core. I know numerous studies argued the cooling history of core based on Fe and Fe-alloy thermal conductivity as shown in Table 1 and 2, but I think that they are useless papers.

I have several comments on the presentation.

7. Although the abstract states "15 at% Si alloyed in Fe substantially reduces the conductivity by about 2 folds at 132 GPa and 3000 K.", there is no such argument in the main text.

8. The authors used single-crystal and polycrystalline samples. They used both samples for measurements at ambient temperature. However, it is not shown in Fig 1 which sample was used. Furthermore, it is unclear whether single-crystal samples were used for high-temperature measurement. They also should show which sample was used in each measurement in Table S1-S3.

9. Page 10. I do not understand why there is a description regarding MORB.

10. Page 14. It is clear that high-temperature measurements should be much more important than ambient-pressure measurements in view of geophysics. Therefore, the descriptions regarding high-temperature measurement should have larger amounts than those regarding ambient-pressure measurements. I wonder what fitting parameters were used for data analyses of high-temperature measurements. The authors should provide a similar table to Table S4 for the high-temperature measurements.

11. Page 28. I am afraid that Figure S3 is not used anywhere in the text.

12. Page 32. The structures of these tables are confusing. One row has six columns and each of three columns describes one measurement. Even though space-consuming, each row should describe only one measurement.

As seen from the comments above, I am not supportive to this article. The paper lacks convincing experimental data to argue the thermal conductivity of core materials at high pressures and temperatures. More essentially, thermal conductivity of I also point out that this paper is organized unreasonably. I strongly suggest the authors to reconsider their measurement and arguments, and to reorganize the whole parts of the article.

We thank you and the three reviewers for helpful comments. In what follows, all reviewers' comments are in italics and our point-to-point responses are in normal fonts with blue color. Changes in our revised manuscript are also labeled by blue color. We have added a significant amount of materials and information, which, we believe, have fully addressed the reviewers' concerns and questions.

Report of Referee # 1

1. The most important question is whether the experimental measurements and analysis are sufficient to reliably support the authors' conclusions. The authors describe their efforts to propagate uncertainty through the analysis of their measurements. This is important because the thermal conductivity must be extracted from a modelling procedure. Uncertainties in the model parameters will affect the outcome. The authors suggest that the uncertainty is about 10-20%. I wonder if these error estimates are too low. Figure S4 shows the temperature dependence at various pressures. I would be surprised if the erratic pressure dependence is real, which suggests that these variations are reflection of experimental uncertainty. Still, the results are sufficient to argue for lower thermal conductivity, even with larger experimental uncertainties.

The error bars for our high-pressure and room-temperature TDTR thermal conductivity measurements are about 10–25%, depending on the pressure range, while the high pressure-temperature transient heating (TH) measurements have uncertainties of typically about 15–30%. These uncertainties are indeed mainly derived from standard error propagations in the modelling of the experimental spectra after considering various model input parameters. We have clarified the uncertainty issue by adding the descriptions of the errors in the Fig. 2 caption. Error bars are also plotted accordingly in the Fig. S4. The “erratic pressure dependence” the referee referred to in Fig. S4 is due to the uncertainties of 15–30% as well as experimental temperature range in the high P-T transient heating measurements, which also reflect the state of experimental challenge in this type of study. As the reviewer commented, though there are some inevitable measurement errors, our results are sufficient to indicate a low thermal conductivity of outer core when the presence of a light element such as Si is taken into consideration.

2. There is little doubt that this study will motivate future work, either to reproduce these results or to extend the experiments to other light elements. It seems likely that silicon will be present at some level in the core, so the results should be relevant moving forward. I would encourage the authors to make sure that their error estimates will

stand the test of time.

As we respond to the comment #1 above, we have confirmed our experimental error estimates and clarified the discussions of the experimental uncertainties in the revised manuscript where appropriate.

3. The present study could be published in its current form, although I would encourage the authors to be clear early in the paper that the experiments are carried out on solid samples. This detail emerges once the experiments are described, but it is sufficiently important to mention in the abstract. Allowing for the likely changes due to melting only reinforces the authors' conclusions.

Following the reviewer's suggestion, we have emphasized that the experiments were conducted on solid samples in the abstract. This point, together with the melting effect on thermal conductivity, has been made clearly in the main text where appropriate.

Report of Referee # 2

1. Page 2, First paragraph and Page 6 the first paragraph of discussion: I would like to make some discussions on the mechanism of heat transfer at very high pressure and temperature. Generally heat is transferred by three mechanisms, such as convection heat transfer, thermal conduction which is measured here experimentally, and radiation. I am wondering that the radiation heat transfer may not be negligible and this mechanism might be important and can increase the thermal conductivity significantly in spite of the low thermal conductivity determined here. We need some discussions on the heat transfer mechanism in the core.

There is no radiative heat transport in metals, such as Fe and Fe-rich alloys of this study, where radiation can only propagate to less than 100 nm depth—metals are opaque to thermal radiation. On the other hand, the radiative conductivity can arguably play an important role for the heat transport in the mantle materials (dielectric and semiconductors), which are capable to transfer heat by radiation. Thus, there are only two major mechanisms of heat transfer in the iron alloy core: by conduction and convection. To clarify this general subject, we have added the following sentence in line 67-69 on page 3 of the Introduction:

”There are two major mechanisms of heat transfer, i.e., thermal conduction and convection, in the core, where the thermal radiation mechanism does not effectively transfer heat in metallic Fe and Fe-rich alloys.”

2. Page 5, 2nd paragraph: The authors used two techniques, TDTR at 300 K and TH at

high temperature. However, there is no discussion on the consistency of these technique. I would like to see some discussions on the consistency of the two data-sets. The data on TDTR at room temperature might extrapolate theoretically, and could compare the data by TH. The Si content dependency for TDTR and TH are consistent?

The reviewer raised a good point about comparing and confirming the consistency of TDTR results at high pressure and room temperature with TH results at high pressure and high temperature. To compare and confirm the two TDTR and TH datasets, we have added a new supplementary figure (Fig. S1) to show the temperature effect on the Fe thermal conductivity at high pressures. As we also showed in Fig. 1, the TDTR data for pure hcp-Fe at the outermost core pressures and 300 K is about $120 \text{ W m}^{-1} \text{ K}^{-1}$. If we assume the temperature dependence of the pure hcp-Fe follows a $T^{-1/2}$ dependence as estimated by Konôpková et al. (Nature 2016), the thermal conductivity of hcp-Fe at P-T conditions relevant to the outermost core conditions ($P=136 \text{ GPa}$, $T=3800\text{--}4800 \text{ K}$) is estimated to be about $30\text{--}33.7 \text{ W m}^{-1} \text{ K}^{-1}$, nearly the same as that ($33\pm 7 \text{ W m}^{-1} \text{ K}^{-1}$) obtained by TH measurements in Konôpková et al. (Nature 2016). This comparison thus confirms experimental TDTR and TH results for pure iron.

On the other hand, the exact temperature dependence of the thermal conductivity of Fe-Si alloys at high pressures with different Si content remains relatively uncertain. As we reported in Fig. 2 and Fig. S4, the temperature dependence likely varies with Si contents and applied pressure as well. Thus, without the knowledge of the temperature dependence of Fe-Si alloys at high pressures, it may be difficult to make unambiguous extrapolation of room-temperature TDTR data to high-temperature conditions and compare them with the high-temperature TH data. However, we note that *qualitatively* these two sets of data correspond reasonably well. Both sets of data demonstrate pressure dependencies with a broad maximum for Fe-Si alloys (after about 40 GPa at 300 K and around 80–100 GPa at 2050–3300 K). Moreover, given the Si alloying effect, it is expected that the thermal conductivity of Fe-Si alloys would have weaker temperature dependence than that of pure hcp Fe, since the presence of impurities will enhance the scattering of carries (phonons and electrons) during their transport of energy. This qualitative behavior is clearly indicated in the newly added supplementary Fig. S1-S3.

To discuss the consistency of the TDTR and TH results, we have added a new paragraph (as we stated above) in line 148-165 on page 6-7 of the section “Discussion”. In addition, to add further comment on the effect of temperature on the thermal conductivity, we have also added a sentence in line 178-181 on page 8:

“On the other hand, the variation in high P-T thermal conductivity of Fe-Si alloys is likely due to the P-T effects on electron-impurity scattering contribution to the conductivity (see Fig. S4 and Supplementary Information).”

3. Figure S3: There is no explanation of the blue curve. No explanation for FE (finite-elements) calculation. FE calculations should be FE calculation. Please fix this figure.

We apologize for the omissions. We have now explained the fitting curves by the finite-element calculation in the caption of the new Fig. S9 (originally Fig. S3). The new figure is also cited in the main text. Additional information about the TH technique and data analysis (FE calculation) has also been added onto the Methods. Typos in the figure caption have been corrected.

4. Figure S4: The change of the FeSi thermal conductivity shows strange pressure dependency at high temperature, i.e. there is a maximum at 106 GPa. Provide a reasonable explanation of the pressure dependency at high temperature of 2500-3000 K. Is it due to a two phase mixture of BCC and HCP? If the anomalous change is due to uncertainty of the TH method, please provide uncertainty for the plot for the measurements.

This comment is somewhat related to the comment #2 above. Our results on Fe-Si alloys show that the temperature-dependent thermal conductivity likely changes with Si contents as well as applied pressure. As we have explained in line 137-145 on page 6, the co-existence of BCC phase with HCP phase is not a main contributing factor. To further clarify the P-T and Si effects on the thermal conductivity of $\text{Fe}_{0.85}\text{Si}_{0.15}$ alloy, we have revised Fig. S4 and Fig. 2. We added a red curve (which shows the averaged values of the high-T measurements at each given pressure) in Fig. 2 for the thermal conductivities (red symbols) to show the trend: the thermal conductivity of $\text{Fe}_{0.85}\text{Si}_{0.15}$ reaches a broad maximum around 80–100 GPa, followed by a decrease with higher pressure to 144 GPa. As the reviewer pointed out, such pressure evolution at high temperatures is complementarily revealed in Fig. S4, where we have also added the error bars to illustrate the measurement uncertainty. We explain the pressure-temperature dependence as follows and added a paragraph in the new section “Effects of pressure, temperature, and Si alloying on the thermal conductivity of Fe-Si alloys” in Supplementary Information:

Thermal conductivity of Fe or Fe-rich alloy is dominated by the electrical conductivity. The electrical conductivity (σ) and resistivity (inverse of conductivity) are strong functions of T ($1/\sigma$ is proportional to T). However, the temperature dependence of

thermal conductivity should be weaker as it can be determined via the Wiedemann–Franz (WF) law $k=L\times\sigma\times T$, where k is the thermal conductivity and σ the electrical conductivity, and L the Lorenz number. That is why a small change in the T dependence of resistivity with pressure would result in a change of the T dependence of thermal conductivity, which can increase or decrease with T (Fig. S4). This T dependence of thermal conductivity may also vary with the Si composition, making k decreasing with T (as in Konopkova et al., 2016) or increasing with T (as in this work at 106 GPa, Fig. S4). Please also refer to our responses in the comment #2 above regarding the effect of temperature on the thermal conductivity of Fe-Si alloys.

5. Figure S5: Phase diagram shows Fe 2-3.5wt% Si. In the caption, alloy compositions are given as atomic in composition. Fe 2-3.5wt%Si should change to atomic ration as shown in the caption. It is confusing. The symbols of this figure are not clear. There are blue diamonds and upwards triangle, blue open circles and downwards triangles, and orange open stars. There are no detailed explanations of the symbols. What does the meaning of the top side of the sample and the bottom side of the sample? The top side and bottom side of the sample is related to the pulsed side and opposite side given in Figure S3?

We have changed the label from wt% to “Fe 4-7 at% Si” in Fig. S5. We have also added explanations for each of the symbol used in the revised Fig. S5 caption: The symbol at the top (bottom) of each pressure represents the high P-T measurement conditions taken from the pulsed (probe) side of the sample. For instance, in the revised Fig. S5, the dark brown up-triangles represent the P - T conditions for $\text{Fe}_{0.96}\text{Si}_{0.04}$, where the temperatures were measured from the pulsed side of the sample, while the blue stars represent the P - T conditions for $\text{Fe}_{0.93}\text{Si}_{0.07}$, where the temperatures were measured from the probe side of the sample. The reviewer is correct that “the top side is the pulsed side and the bottom side is the opposite side of the sample” in our original submitted manuscript. To avoid confusion, we have changed the “top side” to “pulsed side” and “bottom side” to “probe side” throughout the revised manuscript.

6. Figure S6: Provide a reference for the phase diagram of FeSi alloy. It may be ref. 12, but it is confusing that black triangles and open circles are the literature data for phase identification by ref. 12, but NOT the data points for previous thermal or electrical conductivity measurements. The measurement made with TH experiments was for the mixture of hcp and bcc, and not a single phase. Please explain the differences of your data points shown in yellow stars and brown stars. Do you show the two series of measurements? What is “Bottom temperature” in this figure? The solid lines for the

boundaries of bcc, hcp, fcc may be those of pure Fe instead of Fe_{0.85}Si_{0.15}. Therefore it should be deleted.

The phase diagram in Fig. S6 is from Lin et al. (Ref 12)—the reference, Lin et al.¹², has been provided in the original version of the manuscript. In the revised Fig. S6 caption, we have clarified the meaning of the symbols:

Dark yellow (brown) stars are the P-T conditions of our high-temperature TH experiments collected from the pulsed (probe) side of the sample. These symbols represent the range of sample temperature variation measured by radiative temperature measurements from pulsed (probe) side of the sample.

Thus the original “bottom temperature” meant the temperature measured at the probe side of the sample, represented by the brown stars. Similar to the symbol issues in Fig. S5 in the comment #5, to avoid confusion, we have changed the “top temperature” to “pulsed side” and “bottom temperature” to “probe side” in the revised Fig. S6 and caption. Following the reviewer’s suggestion, we have also removed the phase boundaries (solid curves) for pure Fe and literature data of black triangles and open circles.

Report of Referee # 3

1. I think that their main conclusion that the Fe-Si alloys have low thermal conductivity than pure Fe is invalid at high pressures and temperatures. Figure 2 demonstrates that the thermal conductivities of pure Fe and Fe-Si alloys are simply indistinguishable at high temperatures, although it may be the case at ambient temperature as shown in Fig. 1.

We are afraid that the reviewer may have misunderstood our results in high P-T and Si effects on the thermal conductivity. To clarify high P, T, and Si effects separately, we have added Figures S1 for thermal conductivity of pure iron, S2 for Fe_{0.96}Si_{0.04} alloy, and S3 for Fe_{0.85}Si_{0.15} alloy at high pressure as well as high P-T. Together with Figures 1 and 2, our results clearly show a very low thermal conductivity of Fe_{0.85}Si_{0.15} at relevant outer core conditions, such as 132 GPa and 144 GPa (representing the conditions and the composition of the Earth’s outer core), about 16–29 W m⁻¹ K⁻¹ and 16 W m⁻¹ K⁻¹, respectively. These values are much smaller than about 44 W m⁻¹ K⁻¹ for pure iron reported by Konopkova et al. (2016) at similar high P-T conditions (Fig. S1-3); the thermal conductivity difference between Fe_{0.85}Si_{0.15} and pure Fe is well beyond the measurement errors labeled in the Fig. 2, presenting a solid evidence to support our conclusion.

2. As shown in Table S1-S3, the measurements for $Fe_{0.96}Si_{0.04}$ and $Fe_{0.93}Si_{0.07}$ were conducted at a temperature of 2050 K, whereas those for $Fe_{0.85}Si_{0.15}$ were at temperatures of 2300 to 3100 K. Therefore, this study does not present consistent experimental results to show the compositional dependence of thermal conductivity of Fe-Si alloys.

We hope the reviewer can understand that the transient heating (TH) technique requires a temperature gradient from the pulsed laser side to the probe laser side of the sample such that there's always a range of temperature quoted for each given measurement at high pressure and temperature. Performing simultaneous high pressure-temperature thermal conductivity measurements is extremely challenging, limiting our reported pressure-temperature ranges in experimental runs that the referee referred to. Though the temperature conditions are slightly different for different Si contents, the nominal temperature difference here will not significantly change our results on the compositional dependence of thermal conductivity (as demonstrated in Figure 2).

3. The thermal conductivity of Fe and Fe-Si alloys increases by adding percent orders of Si to Fe, whereas it decreases by adding 15at% of Si. This would indicate that the effect of Si addition is monotonic, and therefore, they cannot argue core thermal conductivity simply using that of $Fe_{0.85}Si_{0.15}$ alloy.

We believe the reviewer misinterpreted our results and statement. We never state in the paper that the effect of Si is monotonic as our experimental results on four different samples (Fe and three Fe-Si alloys) don't show such a trend. As we have explained in the text, our high P-T results on these Fe and Fe-Si alloys suggest that different amounts of silicon can have different high P-T effects on the thermal conductivity of Fe-Si alloys—addition of Si decreases the thermal conductivity at high pressure and room temperature. While high T decreases thermal conductivity of pure iron at high P, the temperate effect is much smaller in Fe-Si alloy. For deep Earth and core thermal evolution, as we have clearly stated in the manuscript, geophysical and geochemical observations suggest that *iron alloyed with ≈ 15 at% silicon (≈ 8 wt%)* is a likely candidate composition for the outer core. We thus focus on the interpretation of our $Fe_{0.85}Si_{0.15}$ results for geophysical implications.

4. Figure S4 shows that the thermal conductivity of $Fe_{0.85}Si_{0.15}$ increases with increasing temperature at pressures of 47~106 GPa, whereas it is independent from

temperature at higher pressures. The author need to argue the mechanism to cause these phenomena.

This comment is similar to the comments #2 and #4 by the Reviewer #2 above. We explain it again as follows and added a paragraph in the new section “Effects of pressure, temperature, and Si alloying on the thermal conductivity of Fe-Si alloys” in Supplementary Information:

Thermal conductivity of metals is dominated by the electrical conductivity. The electrical conductivity (σ) and resistivity (inverse of conductivity) are strong functions of T ($1/\sigma$ is proportional to T). However, the temperature dependence of thermal conductivity should be weaker as it can be determined via the Wiedemann–Franz (WF) law $k=L\times\sigma\times T$, where k is the thermal conductivity and σ the electrical conductivity, and L the Lorenz number. That is why a small change in the T dependence of resistivity with pressure would result in a change of the T dependence of thermal conductivity, which can increase or decrease with T (Fig. S4). This T dependence of thermal conductivity may vary with the Si composition, making k decreasing with T (as in Konopkova et al 2016) or increasing (as in this work at 106 GPa, Fig. S4). However, we note that, given the Si alloying effect, it is expected that the Fe-Si alloys would have weaker temperature dependences than the pure hcp Fe, since the presence of impurities will enhance the scattering of carries (phonons and electrons) during their transport of energy. This qualitative behavior is clearly indicated in our data of Figs. 1 and 2. (Please also refer to our responses to comments #2 and #4 by the Reviewer #2 above.)

To add further comment on the effect of temperature on the thermal conductivity, we have also added a sentence in line 178-181 on page 8:

“On the other hand, the variation in high P-T thermal conductivity of Fe-Si alloys is likely due to the P-T effects on electron-impurity scattering contribution to the conductivity (see Fig. S4 and Supplementary Information).”

5. The outer core temperatures are more than 4000 K. The present temperature range is insufficient for arguing the core conductivity, because the temperature effect on thermal conductivity does not seem simple as argued above.

We hope the reviewer can understand that thermal conductivity measurements at simultaneous high pressure-temperature conditions are extremely challenging. Though the temperature conditions in our measurement is only up to 3300 K, our results have clearly demonstrated that the thermal conductivity of potential relevant composition in the outer core, $\text{Fe}_{0.85}\text{Si}_{0.15}$, is much lower than previously thought and pure Fe. This key conclusion is expected to remain valid at higher temperatures of about 4000 K because

the temperature dependence of thermal conductivity is expected to be small at the temperature conditions of the outer core (the results at 132 and 144 GPa of Fig. S4).

6. I doubt that argument of the core thermal history based on thermal conductivity of core materials is essential. The core has cooled due to heat release across the CMB. If there were no heat production in the mantle, the released heat should be equal to the total surface heat flow. Thus, the core cooling rate should be controlled by the surface heat flow of the Earth not by the heat transfer in the core. This argument is strengthened by the fact that the thermal conductivity of rocks is much lower than that of Fe alloys. Therefore, it is the thermal-boundary-layer structure of the bottom of the Earth that controls the amounts of heat released from the core to the mantle. The amounts of heat transfer within the core should be adjusted by verbosity of thermal convection in the outer core. I know numerous studies argued the cooling history of core based on Fe and Fe-alloy thermal conductivity as shown in Table 1 and 2, but I think that they are useless papers.

We are afraid that the reviewer's point of view about the relationship between the core thermal conductivity and thermal history is not precise. The reviewer is partially right, in principle, but wrong in the conclusions s/he gets from it. It is right that the mantle controls how much heat goes out of the core. However, even without heat production in the mantle, the CMB heat flow would not match the surface heat flow, unless the mantle is in steady state (i.e., not cooling down), which it isn't. Since the production of radiogenic heating in the mantle and the mantle cooling rate are far from being perfectly pinned down (and even more so for ancient eras), it is useful to find additional constraints on the problem. The persistence of the geomagnetic field for at least 3.5 Gyr is such a constraint that places minimum bounds on the CMB heat flow, and these bounds depend on the thermal conductivity of the core. The CMB heat flow could still be higher than these bounds and this point was made clear in the paper. The problem with the high conductivity values is that the minimum bounds that result from them are almost too high to be acceptable, although we can make it work, e.g. with extraction of light elements from the top. The low values proposed by our paper remove that difficulty. Therefore, determination of thermal conductivity of core Fe alloy at high P-T provides additional constraints on the thermal state and geodynamo of the planet. Much of the information on the subject has been explained in the text in line 35-53.

7. Although the abstract states "15 at% Si alloyed in Fe substantially reduces the conductivity by about 2 folds at 132 GPa and 3000 K.", there is no such argument in the main text.

We have added “by about 2 folds” in line 217 on page 9 to point out such argument.

8. The authors used single-crystal and polycrystalline samples. They used both samples for measurements at ambient temperature. However, it is not shown in Fig 1 which sample was used. Furthermore, it is unclear whether single-crystal samples were used for high-temperature measurement. They also should show which sample was used in each measurement in Table S1-S3.

We clarify that in our room-temperature TDTR measurements, the pure Fe samples include both single-crystal and powder samples, while the $\text{Fe}_{1-x}\text{Si}_x$ ($x=0.04$ and 0.15) alloys are only powder samples. To make the issue clear, we have revised the symbol shapes for pure Fe in Fig. 1 by labeling the powder data as black squares and the single-crystal data as black stars. We also revised the caption accordingly and note the $\text{Fe}_{1-x}\text{Si}_x$ ($x=0.04$ and 0.15) are powder samples.

On the other hand, the high pressure-temperature experiments were performed on polycrystalline (powder) samples, which have been stated in the main text (line 128). To emphasize the sample form in the high temperature measurements, we have added “All the samples for TH measurements are polycrystalline.” in line 282-283 on page 12 of Methods. Thus, the measurements shown in Table S1-S3 are all on polycrystalline samples.

9. Page 10. I do not understand why there is a description regarding MORB.

As we stated in the Methods, MORB is just one of the starting materials to synthesize homogeneous Fe with a specific amount of Si, a part of the synthesis procedure; it has nothing to do with the reported results here.

10. Page 14. It is clear that high-temperature measurements should be much more important than ambient-pressure measurements in view of geophysics. Therefore, the descriptions regarding high-temperature measurement should have larger amounts than those regarding ambient-pressure measurements. I wonder what fitting parameters were used for data analyses of high-temperature measurements. The authors should provide a similar table to Table S4 for the high-temperature measurements.

Following this recommendation, we have added more detailed descriptions of the high P-T experiments in the section “Thermal conductivity measurements at high pressure and high temperature” in Methods. We have also added a new table (Table S5) to list the parameters we used to analyze the data in the new Fig. S9 (originally Fig. S3).

11. Page 28. I am afraid that Figure S3 is not used anywhere in the text.

We are thankful to the reviewer for pointing out this omission. We have substantially extended the description of the high P-T experiments, and now this figure is cited. The new Fig. S9 (originally Fig. S3) shows an example data for the high temperature measurements: temperature evolution of Fe_{0.85}Si_{0.15} foils at 121 GPa during flash heating at high initial temperature.

12. Page 32. The structures of these tables are confusing. One row has six columns and each of three columns describes one measurement. Even though space-consuming, each row should describe only one measurement.

We have added the borderlines to separate the pressure, temperature, and thermal conductivity for each of the measurements in Table S1-S3. We thank the reviewer for pointing this out.

REVIEWERS' COMMENTS:

Reviewer #1 (Remarks to the Author):

The study of thermal conductivity of iron alloy is an important topic for the thermal evolution of the planet and the generation of a magnetic field. The authors' experiment point to much lower thermal conductivity for relevant alloys, compared with previous theoretical calculations or experiments based on measurement of electrical resistance.

The authors have done a good job responding to the reviewers comments. The experiments are carefully done and well explained. This will certainly help other researchers attempt to reproduce these results and consider other possible alloys. Some of the specific predictions likely depend on the choice of alloy elements, but the authors choice is reasonable and the conclusions they draw appear to be appropriate for their specific choice.

I think the paper can published in its current form. It will be of wide interest and generate consider (constructive) debate.

Reviewer #2 (Remarks to the Author):

This manuscript provides the new data on thermal conductivity of Fe-Si alloys at high pressure and temperature, and showed the thermal conductivity of the alloy is very low compared to pure hcp-iron. I have checked carefully the revised manuscript. The authors addressed the reviewers' comments and the manuscript was improved significantly. This low thermal conductivity of the Fe-Si core provides very important implications for older formation of the inner core, energy source of geodynamo, early initiation of the Earth's magnetic field, and small heat flow at CMB. The results and implications are very significant, therefore this manuscript should be published in this journal.

Eiji Ohtani

Reviewer #3 (Remarks to the Author):

This is the second review of this paper. Hence, I do not think that any preface is necessary in this review.

Although the authors claimed that the thermal conductivity difference between Fe_{0.85}Si_{0.15} and pure Fe is well beyond the measurement errors, I disagree to their idea. The values obtained by the present method have actual uncertainties larger than a factor 2. This fact is suggested by the data points at pressures between 100 and 110 GPa, which range from 28 to 60 W m⁻¹ K⁻¹. If the error bars are taken into account, the actual uncertainty is a factor 4 (20 ~80 W m⁻¹ K⁻¹). Although low conductivity values (16-30 W m⁻¹ K⁻¹) were obtained at pressures above 120 GPa, these data should have the same level of errors. This should be also the case with Konopkova et al.'s study. A reasonable conclusion of the present study is therefore "the conductivity of Fe_{0.85}Si_{0.15} is indistinguishable from that of pure Fe within the errors".

Although I appreciate the authors' tremendous efforts for this pioneering study, the robustness should not be exaggerated. If the decrease in conductivity above 100 GPa of Fe_{0.85}Si_{0.15} given in this study were the case, the conductivity of pure Fe given by Konopkova et al. should suggest that the conductivity decreases by a factor of 2 with increasing pressure from 40 to 70 GPa, and suddenly increase by a factor of 2.5 from 70 to 90 GPa, and then remain constant at higher pressures. These

arguments based on Konopkova et al.' data are clearly too much, and therefore the authors conclusion regarding the difference between pure Fe and $\text{Fe}_{0.85}\text{Si}_{0.15}$ is also too much.

I am afraid that high-pressure experimentalists tend to exaggerate the robustness of their studies. If a data set with the same quality is obtained by other methods than high-pressure experiments, no one would say that it is robust.

I think that the present data set is worthwhile to publish in some journal. However, any geophysical argument is too early. The present study should be published at some journal of mineralogy such as American Mineralogist and Physics and Chemistry of Minerals. The authors should argue the thermal history of the Earth's core after being able to obtain that are more robust.

For these reasons, I do not recommend this paper for publication at Nature Communications.

We thank you and the three reviewers very much for helpful comments. In particular, we appreciate that the final reports by Referee # 1 and # 2 are both very positive. In what follows, the remaining concerns and comments by Referee # 3 are in italics and our point-to-point responses are in blue. Changes in our revised manuscript are also highlighted in blue with tracked changes. Following your suggestion, we have toned down our data interpretations and made clear to the readers why the literature data that we took for comparison are rather scattered within a specific pressure range. Please see our revisions in the section “Thermal conductivity at high pressure-temperature conditions” (line 132-138 on page 6) of “Results” in the revised manuscript.

Report of Referee # 3

Although the authors claimed that the thermal conductivity difference between Fe_{0.85}Si_{0.15} and pure Fe is well beyond the measurement errors, I disagree to their idea. The values obtained by the present method have actual uncertainties larger than a factor 2. This fact is suggested by the data points at pressures between 100 and 110 GPa, which range from 28 to 60 W m⁻¹ K⁻¹. If the error bars are taken into account, the actual uncertainty is a factor 4 (20 ~80 W m⁻¹ K⁻¹).

We should note that the data at 100–110 GPa the reviewer referred to actually have the largest uncertainties in our measurements, so this is an extreme case in this study. In comparison, the Fe_{0.85}Si_{0.15} data above 120 GPa have relatively small uncertainties. The data uncertainties presented in Fig. 2 are somewhat scattered and reflect the current status of thermal conductivity measurements at extreme pressure-temperature conditions. Even with these uncertainties considered, the overall data trend clearly show that thermal conductivity of hcp-Fe_{0.85}Si_{0.15} alloy is lower than that of pure hcp-Fe at relevant pressures of the core such as 132 and 144 GPa. Therefore, our geophysical implications on lower thermal conductivity of Fe-Si alloy in Earth’s core are supported by the data.

Although low conductivity values (16-30 W m⁻¹ K⁻¹) were obtained at pressures above 120 GPa, these data should have the same level of errors. This should be also the case with Konopkova et al.’s study.

The data uncertainties in this study are derived from standard error propagation procedures, including averages from multiple measurements and error propagations in data modelling. Thus, the uncertainties also depend on the experimental data quality and are not necessarily the same in each experimental run.

A reasonable conclusion of the present study is therefore “the conductivity of Fe_{0.85}Si_{0.15} is indistinguishable from that of pure Fe within the errors”.

We are afraid that the reviewer seems to mix up the interpretations of Fig. 2 data at different pressure regime. We believe that the results speak for themselves as we argue above. Below about 100 GPa, as the reviewer pointed out, the overall thermal conductivity of Fe_{0.85}Si_{0.15} seems to be only slightly smaller than that of pure Fe, though their difference is comparable to the errors. However, the results above 120 GPa, where our implications for core thermal evolution focus on, quite clearly support low thermal conductivity of hcp-Fe_{0.85}Si_{0.15} than pure Fe by Konopkova *et al.* That is, the results support the notion that silicon alloying decreases thermal conductivity of iron at relevant Earth’s core pressure-temperature conditions.

Although I appreciate the authors’ tremendous efforts for this pioneering study, the robustness should not be exaggerated.

We thank the reviewer for pointing this out. We certainly never intend to exaggerate the data interpretations. Please note that our claims refer mainly to the behavior above 120 GPa which is of relevance for the Earth’s core.

*If the decrease in conductivity above 100 GPa of Fe_{0.85}Si_{0.15} given in this study were the case, the conductivity of pure Fe given by Konopkova *et al.* should suggest that the conductivity decreases by a factor of 2 with increasing pressure from 40 to 70 GPa, and suddenly increase by a factor of 2.5 from 70 to 90 GPa, and then remain constant at higher pressures. These arguments based on Konopkova *et al.*’ data are clearly too much,*

We think that the reviewer misunderstood the results by Konopkova *et al.* and is trying to make comparison *incomparably* here. Konopkova *et al.* indeed reported a large scatter in their data around 40–90 GPa and these were assigned to be partially associated with the γ phase which could affect some of their results more than others (please see Ref. 20 for details); however, γ -Fe disappears above 100 GPa, so the data scatter less in this regime. There is nothing too much here—just challenging experiments.

To make clear why the literature pure Fe data that we took for comparison are rather scattered within a specific pressure range, we have added the following sentence in the section “Thermal conductivity at high pressure-temperature conditions” (line 132-135 on page 6) of the revised manuscript:

“We should note that the large scatter in the literature pure Fe data²⁰ around 40–90 GPa was assigned to be partially associated with the presence of γ phase which could affect

some of its results at high P - T conditions; however, the γ -Fe disappears above 100 GPa, so the data scatter less in this regime.”

and therefore the authors conclusion regarding the difference between pure Fe and Fe_{0.85}Si_{0.15} is also too much.

With all respect, we see not much logic in reviewer’s conclusion here. The Fe-Si system has different phase relations at high P - T conditions, see Supplementary Fig. 6, and their thermal conductivities at core pressure-temperature conditions are not experimentally measured before. Because silicon is a likely constituent light element in the core, our new results on thermal conductivity of Fe-Si alloy have more direct implications to our understanding of the geodynamo and thermal history of the core.

I am afraid that high-pressure experimentalists tend to exaggerate the robustness of their studies. If a data set with the same quality is obtained by other methods than high-pressure experiments, no one would say that it is robust.

We respect this opinion and, as we mentioned above, we certainly never intend to exaggerate our data interpretations. We have toned down our statements concerning the thermal conductivity of Fe and Fe-Si alloys in the revised manuscript by adding and revising the following sentences in the section “Thermal conductivity at high pressure-temperature conditions” (line 132-138 on page 6) of the revised manuscript:

“We should note that the large scatter in the literature pure Fe data²⁰ around 40–90 GPa was assigned to be partially associated with the presence of γ phase which could affect some of its results at high P - T conditions; however, the γ -Fe disappears above 100 GPa, so the data scatter less in this regime. The thermal conductivity of Fe_{0.85}Si_{0.15} (red circles), on the other hand, is slightly smaller than the pure hcp-Fe below 100 GPa, though their differences are within uncertainties. Importantly, the thermal conductivity of Fe_{0.85}Si_{0.15} decreases significantly with increasing pressure from ~120 to 144 GPa.”

I think that the present data set is worthwhile to publish in some journal. However, any geophysical argument is too early. The present study should be published at some journal of mineralogy such as American Mineralogist and Physics and Chemistry of Minerals. The authors should argue the thermal history of the Earth’s core after being able to obtain that are more robust. For these reasons, I do not recommend this paper for publication at Nature Communications.

We are confused by the reviewer’s logic here. If a scientific result were incorrect or misinterpreted, it is not supposed to be recommended for publication, no matter the impact that journal has. We argue that our results about the low thermal conductivity of Fe_{0.85}Si_{0.15} at relevant P - T conditions in the core is beyond any reasonable doubt.

Therefore, our discussions on the thermal history of the Earth's core are supported by the data. We believe our work is suitable to be published in Nature Communications.